# Osteoimmunomodulatory Nanoparticles for Bone Regeneration

**DOI:** 10.3390/nano13040692

**Published:** 2023-02-10

**Authors:** Jingyi Wen, Donglin Cai, Wendong Gao, Ruiying He, Yulin Li, Yinghong Zhou, Travis Klein, Lan Xiao, Yin Xiao

**Affiliations:** 1School of Mechanical, Medical and Process Engineering, Centre for Biomedical Technologies, Queensland University of Technology, Brisbane, QLD 4059, Australia; 2School of Medicine and Dentistry, Menzies Health Institute Queensland, Griffith University, Southport, QLD 4222, Australia; 3College of Chemistry and Chemical Engineering, Hubei University, Wuhan 430061, China; 4The Key Laboratory for Ultrafine Materials of Ministry of Education, State Key Laboratory of Bioreactor Engineering, Engineering Research Center for Biomedical Materials of Ministry of Education, School of Materials Science and Engineering, East China University of Science and Technology, Shanghai 200231, China; 5School of Dentistry, The University of Queensland, Herston, QLD 4006, Australia; 6Australia-China Centre for Tissue Engineering and Regenerative Medicine, Queensland University of Technology, Brisbane, QLD 4000, Australia

**Keywords:** nanoparticles, bone regeneration, osteoimmunomodulation, targeted drug delivery, nanomedicine

## Abstract

Treatment of large bone fractures remains a challenge for orthopedists. Bone regeneration is a complex process that includes skeletal cells such as osteoblasts, osteoclasts, and immune cells to regulate bone formation and resorption. Osteoimmunology, studying this complicated process, has recently been used to develop biomaterials for advanced bone regeneration. Ideally, a biomaterial shall enable a timely switch from early stage inflammatory (to recruit osteogenic progenitor cells) to later-stage anti-inflammatory (to promote differentiation and terminal osteogenic mineralization and model the microstructure of bone tissue) in immune cells, especially the M1-to-M2 phenotype switch in macrophage populations, for bone regeneration. Nanoparticle (NP)-based advanced drug delivery systems can enable the controlled release of therapeutic reagents and the delivery of therapeutics into specific cell types, thereby benefiting bone regeneration through osteoimmunomodulation. In this review, we briefly describe the significance of osteoimmunology in bone regeneration, the advancement of NP-based approaches for bone regeneration, and the application of NPs in macrophage-targeting drug delivery for advanced osteoimmunomodulation.

## 1. Introduction

Treatments for large bone defects caused by cancer, trauma, infection, and progressive congenital conditions remain challenging for orthopedic surgeons [1,2]. Trauma or disease can cause segmental bone defects, a common and severe clinical condition that can delay the union or non-union of bone [3]. Bone grafting is among the most often utilized surgical approaches to treat bone defects; with almost two million annual surgeries, it is the second most frequent medical procedure worldwide following blood transfusion [4]. Despite the availability of grafts, autologous bone is still the preferred option and gold standard because autologous bone grafts have natural osseointegration, osteoinductivity, and excellent biocompatibility. However, appropriate bone tissue for autologous grafting is generally in short supply, and its harvesting is frequently linked with recipient morbidity [5,6]. Alternatively, bone allografts are the second most popular choice for orthopedic treatment, which have provided feasible alternatives for some complicated bone defects without some of the weaknesses of autografts [7,8]. Bone allografts are mainly osteoconductive, with only demineralized bone matrix (DBM) preparations retaining lower osteoinductivity. Despite this, inferior recovery was detected compared with autologous grafts, and the risk of disease transmission and other infectious agents was also documented [9]. More critically, the typical amounts of naturally available bone graft substitutes are still insufficient to meet therapeutic demands, especially in light of the approaching aging and obesity situations worldwide [10]. Such cases call for an urgent need for artificial bone substitutes.

Biomaterials, especially nanoscale materials with high biocompatibility and plasticity, have been widely utilized in preclinical studies for managing bone-associated diseases. Nanomaterials have shown their potential in guided bone regeneration (GBR) and achieved satisfying biocompatibility, mechanical properties, essential barrier function, and enhanced osteogenesis and angiogenesis [11,12]. Recent studies suggest that the immune microenvironment is critical for biomaterial-regulated bone regeneration. The implanted cells or scaffolds often fail to integrate successfully with the host tissues due to the unfavorable immune response. On the contrary, a biomaterial capable of generating an ideal immune environment for osteogenesis benefits bone regeneration, an effect termed “osteoimmunomodulation”. Meanwhile, nanomaterials, especially nanoparticles (NPs), are well-developed in the drug delivery field for multiple disease treatments, which can load and release functional chemicals and proteins to regulate the local immune microenvironments [13]. Multifunctional NPs encapsulated in cell membranes with a wide range of functions are considered as a future-proof platform for targeted drug delivery [14]. Therefore, novel osteoimmunomodulatory nanomaterials are expected to enhance osteoinduction by generating a favorable bone regeneration environment. In this review, we focus on the importance of osteoimmunology in bone regeneration, summarize the effects of using different materials and different modified NPs to further enhance and promote bone regeneration, and discuss the potential application of NPs as osteoimmunomodulatory tools to improve bone regeneration. Primarily, it innovatively focuses on the recent advances in the development of macrophage-targeted nanotherapeutic agents, a novel and popular research field in Material Science and Nanotechnology, pointing out the potential application of this technology in bone healing, and therefore shedding light on future nanomaterial development for advanced osteoimmunomodulation.

## 2. Bone Regeneration Process

Bone regeneration is a complex, well-coordinated physiological process (Figure 1) [15]. Immediately after fracture, the blood vessels which supply blood to the bone are ruptured, resulting in the formation of a hematoma around the fracture site [1]. This hematoma serves as a temporary framework for healing [1]. Inflammatory cytokines such as interleukins (e.g., IL-1), bone-morphogenetic proteins (BMPs), and tumor necrosis factor-alpha (TNF-α) are released into the injury site. These cytokines attract monocytes, lymphocytes, and macrophages, which work together to eliminate dented, necrotic tissue and produce growth factors such as vascular endothelial growth factors (VEGF) to promote angiogenesis for bone healing. Inside the hematoma, granulation tissue begins to develop. More osteoprogenitor cells/mesenchymal stem cells (MSCs) are attracted to the region, where they start to differentiate into chondroblasts and fibroblasts. As a result, chondrogenesis occurs, a collagen-rich fibrocartilaginous network spans the fracture sites, and hyaline cartilage encloses it. Alongside the periosteal layer, osteoprogenitor cells simultaneously construct a surface of woven bone [16]. Osteocytes, osteoclasts, and chondroblasts are typically stimulated to differentiate during endochondral ossification of the cartilaginous callus. The callus of cartilage is trapped and begins to calcify [16]. Subperiosteally, woven bone is deposited. At the same time, newly formed blood vessels grow, allowing MSCs to migrate. At the end of this process, an abrasive callus of immature bone forms. In a process known as “coupled remodeling”, the osteoclasts repeatedly remodel the hard callus [16]. This process involves both osteoblast bone formation and osteoclast resorption [16]. The spongy bone of the soft callus is supplanted by lamellar bone, and the callus center is substituted mainly by compact bone [16]. The vasculature has undergone significant remodeling in addition to these modifications [17].

Numerous essential molecules that control the intricate physiological process of bone regeneration have been identified. BMPs are potent and effective osteoinductive factors that have received the most attention. They promote the differentiation of osteoprogenitors into osteoblasts by encouraging their mitogenesis. BMPs, which act as strong osteoinductive constituents in diverse tissue-engineering products, show much promise for clinical cartilage and bone regeneration [18]. Cervical fusion, the repair of lengthy bone deformities, and craniofacial and periodontitis applications are just a few of the current clinical applications. The US FDA recently approved BMP-7 and BMP-2 for specific clinical conditions, which can be administered in absorbable collagen, food, and drugs [19]. Except for BMPs, biological substances such as growth factors derived from platelets (PDGF) and plasma rich in platelets (PRP), have been found to aid in the healing of bone defects [20].

## 3. Osteoimmunology in Bone Regeneration

Osteoimmunology is defined as the study of the communication between the immune system and skeletal system [1,21]. The skeletal and immune systems appear separate but are integral and closely related [1,22]. The basic framework for immune system regulation is established by the enrichment and different environment provided by bone marrow for the growth of hematopoietic stem cells (HSCs), which are the common progenitors of all immune cell types [22]. The communication between immune and skeletal cells, on the other hand, is critical for the pathogenesis and progression of skeletal damage diseases, postponed bone regeneration, and some other infectious diseases. Osteoclasts, osteoblasts, and immune cells, such as macrophages and T cells, play a crucial role in bone regeneration and healing. They interact with each other and the surrounding microenvironment to regulate bone remodeling balance and determine bone regeneration (Figure 2). As a result, cells from both the immunologic and skeletal systems interfere widely in the same bone microenvironment [22]. The receptor activator of nuclear factor-B (RANK) and RANK ligand (RANKL) osteoprotegerin (OPG) regulates bone homeostasis and the progression of autoimmune bone diseases by recognizing key signals which regulate intercellular communication among bone and immune cells [21]. To initiate differentiation and stimulation programs, RANK present on the surface of osteoclast progenitors should bind to RANKL present on the surface of many other cells (including osteoblasts) inside the bone microenvironment. On the other hand, the activating threshold of the RANK–RANKL axis is influenced by the relative expression of OPG, which intervenes with the RANK–RANKL axis by acting as a coreceptor for RANK. This axis also exists in immune–skeleton interplay, where immune cells can produce RANKL to activate osteoclastogenesis [23]. Importantly, this invention has resulted in the effective treatment of bone loss related to metastasis and osteoporosis, in which RANKL is targeted with a therapeutic neutralizing antibody [24].

In bone injury, immune cells are the first responders at the defect site, restoring vasculature and initiating signal cascades to attract cells to undertake the healing process. T lymphocytes and B lymphocytes are observed at the injury site after three days of injury, and their quantities are diminished when chondrogenesis starts. It has been discovered that T-cell depletion reduces bone health and fracture healing [25]. B lymphocytes are reported to be increased in the injury site and peripheral blood during fracture healing, and reduced production of IL-10 by B cells has been linked to delayed fracture healing. One of the earliest cell types infiltrated in bone healing hematoma is the macrophage, which remains active through the healing process. Derived from the mononuclear phagocyte system (MPS) in the bone marrow, macrophages appear to serve as regulators for the differentiation and function of osteoblasts and osteoclasts, participating in intermodulation as well as interaction to reach equilibrium in bone remodeling, which makes them crucial for bone formation and remodeling [26]. Macrophages have been broadly characterized into unpolarized M0, pro-inflammatory M1 phenotypes (M1a and M1b), and anti-inflammatory M2 phenotypes (M2a, M2b, and M2c) based on local stimulators, surface markers, and different functions (Figure 3) [27]. The M1 macrophages, which can be stimulated by lipopolysaccharide (LPS), interferon-gamma (IFN-γ), or cytokines, including tumor necrosis factor-alpha (TNF-α), primarily infiltrate the site of the bone defects during the early inflammatory stage. In contrast, the M2 macrophages are stimulated by cytokines such as IL-4 and IL-13, which appear during the subacute phase [1]. The function of M1 macrophages includes clearance of intracellular pathogens and secreting pro-inflammatory cytokines, whereas the activation of the M2 phenotype mainly results in anti-inflammatory responses and subsequent tissue healing. Therefore, the M1 phenotype is traditionally considered to induce/enhance inflammation. In contrast, the M2 phenotype can reduce inflammation and promote tissue repair [28,29]. However, some recent researchers have discovered that the presence of M1 macrophages enhances osteogenesis [30], and an excessive exchange to the M2 phenotype leads to fibrous tissue healing [31,32]. Therefore, it is hypothesized that both M1 and M2 are crucial during the bone healing process [1]. During the first stage of healing, the recruited macrophages polarize to pro-inflammatory M1 phenotypes and generally remain at the site of the defect for three–four days, recruiting immune cells and MSCs. Then, they gradually polarize to anti-inflammatory M2 phenotypes along with the healing process, releasing anti-inflammatory cytokines, eliminating inflammation, and promoting tissue restoration [1,33]. Therefore, early and short-term activation of M1 macrophages is essential, as the M1 macrophage depletion or over-inhibition during the initial stages would inhibit tissue healing [34]. Meanwhile, early activation of the M2 macrophages impairs tissue healing and induces fibrous encapsulation. Therefore, it is indispensable to effectively control M1 to M2 polarization at an appropriate time, conduct an osteogenesis-favoring cytokine release pattern, and benefit the subsequent bone formation 2/8/2023 1:04:00 PM.

## 4. Bioapplication of Nanoparticles

Biomaterials, including polymers, ceramics, and metals, are usually utilized in bone regeneration treatments, which act as bone substitutes or tissue engineering scaffolds [36]. Biomaterials for bone-associated applications have undergone significant improvement in recent years, intending to generate functionalized materials capable of delivering bioactive chemicals that may directly regulate cell activity [37]. The anatomical intricacy of bone makes bone one-of-a-kind and nearly impossible to replicate in artificial materials, along with the severe mechanical stress to which it is subjected. Nonetheless, certain tactics have been implemented with success [38] via nanotechnology. Nanotechnology has enabled the creation of nanostructures to mimic the structures and sizes found in natural bone. Nanomaterials exhibit unique physical and chemical properties, making them attractive for various applications in various fields, including medicine, electronics, energy, and the environment. The physical and chemical properties of nanomaterials are determined by their size, shape, composition, and surface characteristics. One of the most significant physical properties is their size, which results in a large surface area and enhanced reactivity, making nanomaterials more reactive than their bulk counterparts. The shape of the nanomaterials ranges from spherical, rod-like, or triangular to more complex shapes, which can affect their performance, such as the dispersibility in the liquid base [39]. Chemical properties of nanomaterials, such as composition, surface chemistry, surface charge, solubility, and hydrophobicity/hydrophilicity, can affect their stability, solubility, and reactivity, as well as their interaction with other materials and biological systems. The surface charge of nanomaterials can affect their interaction with other materials and biological systems and can be used to control the release of therapeutic agents [40]. Structural properties determine the size and shape of the nanomaterials and the arrangement of the atoms in the material. For example, the electrical, optical, and magnetic properties of nanomaterials are significantly affected by the performance of atoms in the NP structure [41].

Nanoparticulate systems, bioactive glass, hybrid materials, metal and metal oxide nanomaterials, and carbon-based nanomaterials are categories of osteoimmunomodulatory nanomaterials that have gained significant attention in recent years regarding their potential applications in bone tissue engineering. Nanoparticulate systems, including NPs, liposomes, and dendrimers, have effectively delivered therapeutics to bone tissue [42]. Bioactive glass has osteoinductive and osteoconductive properties, making it valuable for promoting bone growth and repair [43,44]. Hybrid materials combine inorganic and organic materials to enhance biological responses, making them ideal for bone tissue engineering applications [45]. Metal and metal oxide nanomaterials exhibit antibacterial and anti-inflammatory properties, making them useful for preventing infections in bone tissue [46]. Carbon-based nanomaterials, such as graphene and carbon nanotubes, have high mechanical strength and excellent biocompatibility, providing a supportive scaffold for bone cells to grow and proliferate [47]. Different types of NPs and nano-hybrid particles, such as ceramic and metal NPs, are used as material coatings and provide great potential for material modification [48]. Hence, NPs can change the scaffold qualities, resulting in improved attributes such as better mechanical properties, induced osteoinduction, and improved osteoconduction. NPs are prospective biomaterials with sizes smaller than 100 nanometers, which have an essential influence on modern medicine [49] by delivering therapeutics in a controlled and reliant manner [50]. There are two main types of NPs: organic (e.g., liposomes, polymeric NPs) and inorganic NPs such as silica, carbon, magnetic, and metallic NPs (Figure 4).

Liposomes have been used in drug delivery. To achieve drug delivery, the cargo should be included in the liposome structure [51]. Depending on the characteristics of the products to be transported, this process can be carried out in two ways. If the cargo is hydrophobic, it is combined with an organic solvent and incorporated into the hydrophobic portion. However, when the cargo is hydrophilic, it should be supplied as an aqueous medium so that it can be retained in the inner section of the liposome. Liposome size is another critical factor that directly impacts the circulatory period. Liposomes throughout the nanoscale range, in particular, can be used to administrate therapeutics [52]. The major disadvantage of liposome biomedical application is that the reticuloendothelial system can recognize liposomes quickly, which facilitates the removal of liposomes from circulation [53] and impairs their drug delivery efficiency.

Polymers are employed to synthesize polymeric NPs. The self-assembly of adaptive block copolymers could also produce structures with a high degree of complexity. Another benefit of polymeric NPs is their high drug-loading capacity [54]. The loaded molecules can be directly dissolved, distributed, or bonded to polymeric elements through covalent connections. As a result, polymeric NPs are now being used to deliver molecules in various biomedical fields, including vaccination, cancer treatments, inflammation, neurologic diseases, and tissue regeneration [55].

Silica is well-known for its biocompatibility, chemical stability, and well-defined surface features. Silica-based NPs, especially mesoporous silica NPs (MSNPs), have been widely applied due to their adjustable particle and pore size, easy surface modification, specific porous structure, high surface area, big pore volume, etc. Consequently, MSNPs can load immense quantities of biomolecules [56]. For bioapplication, MSNPs with pore sizes ranging from 2 nm to 50 nm are ideal choices [57]. Additionally, MSNPs are resistant to degradation by heat, pH, mechanical forces, and dissolution and are thus ideal drug vehicles. Furthermore, their good biocompatibility, ease of production, and excellent binding to multiple antibiotics suggest the good bioapplication potential of MSNPs [58].

In addition to drug delivery, NPs have attracted significant attention in medical imaging. For example, iron oxide NP-based fluorescent probes have been well-accepted [59]. Meanwhile, the versatility of gold NPs makes them appealing for bioimaging procedures. The optical properties of the AuNPs can be adjusted and optimized by engineering the shape and size ratio of the AuNPs [60]. Tailored to absorption nearly in the infrared range, gold NPs allow for better visualization of the deep tissue [61]. Biological applications, including biosensing and diagnostics, can benefit from this technology [62].

## 5. Application of NPs in Bone Regeneration

As a nanostructured material, bone comprises organic and inorganic components with hierarchical structures ranging from the nano- to the macroscopic level. In addition to traditional treatments, nanomaterials offer a novel strategy for bone repair. Nanostructured scaffolds control cellular proliferation and differentiation, which contributes to the regeneration of healthy tissues, and give cells a more supportive structure comparable to native bone structure [63]. The specific properties of NPs, including their physical properties, chemical properties, and different modifications, as well as their quantum physical mechanisms, make them advantageous over conventional materials [64]. There are plenty of approaches using NPs to regulate bone regeneration. For example, in the initial implantation period, NPs can be an effective enhancer on the surface of biomaterials to acquire good mechanical properties and stability, providing structural function in the injury site for bone healing [65]. NPs can also be incorporated into biomaterials to offer them adjustable mechanical strength (stiffness), stimulating stem cells to take on an extended shape to differentiate preferentially into osteoblasts [66,67]. Meanwhile, a CaP ceramic–magnetic NP (CaP-MNP) composite can use magnetic fields to promote bone healing [68]. Moreover, some NPs themselves can directly improve osteogenesis. For instance, titanium oxide nanotubes of 70 nm diameter induced osteogenic differentiation by regulating H3K4 trimethylation [69]. In the deficiency of any osteoinductive factor, one kind of synthetic silicate nanoplatelet can promote the stem cells’ osteogenic differentiation [70]. Another common application of nanotechnology in bone regeneration is to use NPs to load biomolecules/drugs facilitating osteogenesis, including osteoinductive factors (e.g., osteopontin, BMPs, VEGF) [71,72,73]; drugs reducing bone resorption; and inducing osteogenesis (e.g., alendronate, simvastatin, dexamethasone) [74,75,76], microRNAs (e.g., miR-590-5p, miR-2861, miR-210) [75,77,78] and others [55,79,80].

Despite delivering one bioactive factor, combining two growth factors can better mimic the natural process of bone healing. For example, stromal cell-derived factor 1 (SDF-1), a significant chemokine for stem cell migration, plays a crucial role in the recruitment of MSCs. Meanwhile, BMP-2 is an inducer of osteogenesis in MSCs. Wang et al. introduced a chitosan oligosaccharide/heparin NPs for delivery. They sustained the release of BMP-2 and SDF-1, which sequentially induced migration of MSCs and promoted their osteogenic differentiation for bone repair, an efficient strategy to avoid the rapid degradation of SDF-1 and BMP-2 [81]. Another research study by Poth et al. also loaded BMP-2 on bio-degradable chitosan-tripolyphosphate NPs to induce bone formation [73].

VEGF is a kind of growth factor that plays a vital role in the process of angiogenesis [82]. VEGF is primarily expressed during the early stages to promote blood vessel formation and re-establish vascularization throughout normal bone repair and healing. Meanwhile, BMPs are uninterruptedly expressed to stimulate bone remodeling and regeneration [83,84]. Many researchers have reported that the synergistic effects of BMP-2 and VEGF would better benefit bone regeneration than one growth factor. VEGF expression in bone defects can upregulate the production of BMP-2, which is indispensable in bone healing [85,86]. As a result, more and more studies focused on the co-delivery of VEGF and BMP-2 using NPs. Geuze et al. created poly(lactic-co-glycolic acid) (PLGA) microparticles for sustained release of BMP-2 and VEGF, which achieved improved osteogenesis [84]. Young Park et al. developed 3D polycaprolactone (PCL) structures with hydrogel decorated with both VEGF and BMP-2 and showed more capillary and bone regeneration compared with the delivery of BMP-2 alone [87]. To achieve sequential release of VEGF and BMP-2, some researchers used microspheres (e.g., PLGA microspheres, O-Carboxymethyl chitosan microspheres) loaded with BMP-2 integrated into scaffolds (e.g., poly(propylene) scaffold, hydroxyapatite collagen scaffold) loaded with VEGF. The scaffolds exhibited a substantial initial strong release of VEGF and a sustained release of BMP-2 over the rest of the implantation period. These studies indicated that it is beneficial for bone formation and remodeling to have a sequential angiogenic and osteogenic growth factor secretion [88,89].

Nanoemulsification is one of the most common and well-known methods for producing NPs. It is characterized by synthesizing nanosized particle dispersions by combining the polar phase with the non-polar phase when a surfactant is available and enables the production of 100 nm, injectable, 3D-printable with a high specific surface area and limited mass transport restrictions NPs. Hydroxyapatite NPs synthesized via nanoemulsion technology are thoroughly explored as inorganic components of composite bone implant materials. The combination of nano-hydroxyapatite with an elastic biodegradable polymer, which mimics the organic materials of bone extracellular matrix, has been demonstrated to enhance viability, adhesion, and proliferation significantly. Osteogenic differentiation of cells seeded onto implants such as human mesenchymal stem cells (hMSCs), which is attributed to osteoinductive properties of hydroxyapatite nanomaterials [90]. Additionally, the NPs synthesized from hydroxyapatite and metal materials have significant bactericidal properties [91]. Therefore, nano-hydroxyapatite has been used to create osteoinductive coating materials for bone implants, a strategy to facilitate their osseointegration with the host tissue [92]. Bone implants modified with silver NPs synthesized by bioreduction techniques have enhanced antibacterial and antioxidant properties [93].

Recently, many endeavors have been devoted to developing NPs that bind specifically to the bone. Such NPs can accumulate at the targeted sites, increasing therapeutic efficiency, limiting the adverse side effects of the drug delivery to other tissues/organs [94] and can be widely used in diagnosis, bone tissue engineering, and treatment of bone disease [95]. Bone-targeting NPs are typically created by modifying them with compounds with high affinity for bone tissue, such as Ca^2+^ ions. Examples of these compounds include bisphosphonates (BP), which comprise two Ca^2+^-binding phosphonate groups in their molecules [96], and alendronate, an anti-osteoporotic drug that can bind to hydroxyapatite via multiple Ca^2+^ ions [97]. When NPs are functionalized with alendronate, they can selectively target bone, restraining bone resorption and acting as “anchors” to strengthen the interaction of the implant with the host tissue [98,99]. For this reason, alendronate has been widely utilized for the functionalization of NPs for bone regeneration applications such as inorganic (e.g., Fe_3_O_4,_ hydroxyapatite, clay) [80,100,101] and polymer (e.g., poly(g-benzyl-L-glutamate), PLGA) NPs [55,79,99].

NPs have unique properties, such as a high surface area-to-volume ratio, which can make them more efficient delivery vehicles for drugs and other therapeutic agents. However, their unique properties also raise several safety concerns, primarily related to their biocompatibility, immunogenic properties, and toxicity.

NPs are generally considered biocompatible as long as they do not cause obvious inflammation or irritation. Otherwise, the application of NPs can be limited due to their bio-incompatibility. One study showed that 50 nm-sized particles of Fe_2_O_3_-NP caused severe oxidative stress in HepG2 cells and extreme damage in rat liver [102]. NPs may be immunogenic if they contain foreign proteins or other molecules the body recognizes as threats. Immunogenic NPs can trigger an immune response, leading to inflammation, cell death, and other adverse reactions [103]. The toxicity of NPs depends on their composition and size. Smaller NPs have a larger specific surface area and therefore are more likely to interact with cellular components and are more likely to enter cells and be taken up by organs, which can result in toxicity. For example, in one study, the effects of silver nanoparticles of different sizes (20, 80, 113 nm) on cytotoxicity, inflammation, genotoxicity, and developmental toxicity were compared in in vitro experiments, and 20 nm silver nanoparticles were more toxic than larger nanoparticles [104]. The released Ag^+^ endangers cellular functions, causing damage to deoxyribonucleic acid and cell death [105].

NPs have been frequently used in bone regeneration in recent years. Integrating nanotechnology into tissue engineering applications has created a plethora of new potential for researchers and new clinical applications.

## 6. Applications of NPs in Osteoimmunomodulation

Osteoimmunomodulation refers to the modulation of the immune system to make the local immune environment beneficial for bone regeneration. It aims to use functional materials to regulate the immune cell responses to sequentially modulate the bone remodeling processes, facilitating bone healing [106]. It involves regulating immune cells or cytokines to influence bone remodeling and maintain bone health [107].

Immune suppression benefits certain conditions, such as allergies, autoimmune disorders, and organ transplants. Immunomodulatory or anti-inflammatory characteristics are required for these applications. Several experimental and characterization methods are used to assess the properties of nanomaterials, such as polymers, ceramics, composites, and metals in osteoimmunomodulation (Table 1).

Engineered NPs serve as vehicles for delivering anti-inflammatory drugs to phagocytes, lowering therapeutic doses and immune-related adverse effects [108]. Immune system activation is inevitable when NPs invade. The innate immune cells interact with newly initiated NPs immediately and produce complex immune reactions as a first defense against impending threats to the host. Depending on their physicochemical characteristics, NPs can engage the interactions between proteins and cells to stimulate or inhibit the innate immune response and complement system activation or avoidance. NP size, structure, hydrophobicity, and surface chemistry are the major factors that affect the interactions between the innate immune system and NPs [109].

For bone regeneration, immunomodulation is required to generate an ideal environment for the subsequent osteogenesis, which can be achieved by NPs. As explained in Section 3, macrophage populations are critical regulators of bone regeneration. The pro-inflammatory M1 phenotype of macrophages causes a rise in pro-inflammatory cytokines such as IL-1β, IL-6, and TNF-α, resulting in the inhibition of osteogenesis [110,111] and promoting osteoclastogenesis [112]. Alternatively, the anti-inflammatory M2 phenotype can reverse inflammation and secrete osteogenic cytokines, including BMP2 and VEGF, to encourage bone regeneration [113,114,115]. Hence, targeting macrophages to induce their M2 polarization has been regarded as an efficient way to enhance bone regeneration, and nanomaterials are shown as effective agents for macrophage polarization (Table 2). Some NPs (Figure 5) can efficiently promote M2 polarization, such as gold, TiO_2,_ and cerium oxide (CeO_2_) NPs [116,117,118]. Moreover, the nanopore structure and pore size were discovered to affect the inflammatory response and release of pro-osteogenic factors of macrophages by influencing their spreading, cell shape, and adhesion [119,120]. For instance, Chen et al. ascertained that macrophages grown on larger pore size NPs (100 and 200 nm) were highly anti-inflammatory, demonstrating a decrease in pro-inflammatory cytokine and expression of M1 phenotype surface-marker [119]. One study found that silver NPs with different sizes and shapes showed different effects on bone metabolism and immunity, indicating that controlling the size and shape of nanomaterials can affect their osteoimmunomodulatory effects [121]. NPs with rough surfaces also alter macrophage activation and cytokine release. Research indicated that titanium (Ti) with a smooth surface could induce M1 activation and inflammatory cytokines expression, including IL-1β, IL-6, and TNF-α. Meanwhile, Ti with a rough and hydrophilic surface enhances anti-inflammatory macrophage polarization and the secretion of cytokines such as IL-4 and IL-10 [122]. Another way to promote M2 polarization is to modify the composition of NPs surfaces by doping anti-inflammatory elements or decorating bioactive molecules [123,124,125]. For example, hexapeptides Cys-Leu-Pro-Phe-Phe-Asp [112], peptide arginine-glycine-aspartic acid (RGD) [126], and IL-4 [127] have been successfully conjugated on gold NP surfaces to achieve successful anti-inflammation. Besides, CeO_2_ NPs have been coated with hydroxyapatite to promote M2 polarization [128]. A previous study indicated that surface modification of hydroxyapatite nanorods with chitosan reduced macrophage activation and enhanced osteoblast proliferation [129]. Moreover, strontium (Sr) or copper (Cu)-decorated bioactive glass particles have been found to enhance M2 polarization and promote osteogenesis [124,125]. Zhang et al. synthesized strontium-substituted sub-micron bioactive glasses (Sr-SBG), which have been found to advance the proliferation and osteogenic differentiation of mMSCs [130].

As potential drug delivery systems, NPs have been widely used for bioactive molecule delivery, such as cytokines, growth factors, gene-modulators, and signaling pathway regulators, to stimulate the M1-to-M2 polarization. For instance, IL-4, a widely used anti-inflammatory cytokine, has been frequently adopted as cargo delivered by various nanocarriers to induce M2 polarization [131,132,133]. One research study introduced an IL-4-incorporated nanofibrous heparin-modified gelatin microsphere, which can alleviate chronic inflammation due to diabetes and improve osteogenesis [132]. Sphingosine-1-phosphate (S1P), as a sphingolipid growth factor, can also stimulate macrophages to polarize to the M2 phenotype [134]. Das et al. synthesized nanofibers composed of polycaprolactone (PCL) and poly (D, L-lactide-co-glycolide) (PLGA) for an S1P synthetic analog delivery, which was found to induce macrophage differentiation to M2 phenotypes, facilitating osseous repair in an animal model of the mandibular bone defect [135]. CD163 is an M2 phenotype marker affiliated with the scavenger receptor cysteine-rich (SRCR) family [136]. One study encapsulated CD163 gene plasmid into polyethyleneimine NPs assembled with a mannose ligand for selectively targeting macrophages and inducing CD163 expression, and further transferring macrophages into their anti-inflammatory phenotype [137]. Upregulation of miR-223 can drive the macrophage polarization toward the anti-inflammatory (M2) phenotype, whereas local-targeted delivery of miRNAs is still challenging due to the low stability of miRNA. To solve this problem, Saleh et al. developed an adhesive hydrogel with NPs loaded with miR-223 5p mimic to regulate macrophage polarization to M2 to promote tissue remodeling [138]. Yin et al. loaded an anti-inflammatory drug, resolvin D1, into the gold nanocages (AuNC) coated with cell membranes from LPS-stimulated M1-like macrophages to facilitate M2 polarization. The overexpressed inflammatory cytokine receptors on the cell membrane can competitively bind to the pro-inflammatory cytokines with cell surface receptors, thereby impeding inflammatory responses [139]. The results indicate that this nanosystem could efficiently inhibit inflammatory responses, stimulate an M2-like phenotype polarization, and promote bone regeneration in the femoral defect.

Despite the crucial role of M2 macrophages in promoting bone tissue regeneration, more and more studies have focused on the importance of M1 macrophages in osteoimmunomodulation. As mentioned, M1 macrophages dominate in the early stage of inflammation, enhancing the early commitment and recruitment of angiogenic and osteogenic precursors. In contrast, M2 macrophages function in the later stage of bone regeneration by facilitating osteocyte maturation and determining the microstructure of the newly formed bone tissue [140]. Therefore, a highly orchestrated immune response comprising sequential activation of M1 and M2 macrophages is essential for subsequent bone healing [141]. Thus, a sequential release of therapeutics from NPs to instruct the timely phenotypic switching of macrophages is deemed necessary. For example, as IFN-γ and IL-4 can induce M1 and M2 polarization, Spillar et al. designed a scaffold with a quick release of IFN-γ to increase the M1 phenotype, subsequently with a release of IL-4 to enhance the M2 phenotype. The sequential release feature was achieved by physically adsorbing IFN-γ onto the scaffolds, while loading IL-4 on the material via biotin-streptavidin binding [142]. In another example, miRNA-155 is highly expressed in M1 and less in M2, while the delivery of miRNA-21 can promote macrophage polarization toward M2 phenotypes [143,144,145]. Li et al. synthesized NPs through free radical polymerization carrying both miRNA-155 and miRNA-21 to induce macrophages first toward M1 sequentially and then M2 polarization, a new strategy for bone regeneration [146]. Zinc (Zn) is an essential trace element in various immune responses. Zn’s scarcity and low concentration caused inflammation, while a proper concentration of Zn exhibited an anti-inflammatory effect [147,148]. Therefore, one study fabricated microcrystalline bioactive glass scaffolds with different doses of ZnO to orchestrate the sequential M1-to-M2 macrophage polarization, taking advantage of varying amounts of Zn^2+^ released from the material [149]. Yang et al. incorporated IFN-γ and Sr-substituted nanohydroxyapatite (nano-SrHA) coatings to the surface of native small intestinal submucosa (SIS) membrane, which is widely applied in GBR to direct a sequential M1-M2 macrophage polarization. The nano-SrHA coatings were loaded on the SIS membrane using the sol-gel method, while the IFN-γ was physically deposited. As a result, the physically absorbed IFN-γ released in a burst manner to induce temporary M1 macrophage polarization, then a more sequential release of Sr irons to promote M2 polarization, which intensely improved the vascularization and bone regeneration [150]. Bone marrow macrophages have various receptors on their surface that enable them to recognize molecules such as cytokines, chemokines, lipids, and glycans. NPs to ensure a drug delivery to target bone marrow macrophages can be achieved using strategies such as surface modification of NPs with components interacting with bone marrow macrophage receptors. However, NPs in circulation are removed by the mononuclear phagocyte system (MPS), including the spleen, liver, and Kupffer cells, affecting the NP-based targeted delivery on bone marrow macrophages. Therefore, combining the NPs with bone implants (via approaches such as surface coating, 3D printing, etc.) is suggested instead of systemic administration, which can facilitate the NPs to modulate the local bone healing immune environment and avoid particle clearance due to blood circulation and MPS.

Taken together (Figure 6), osteoimmunology is a fascinating field focusing on the interconnected molecular pathways between the immune and skeletal systems. Among all the immune cells, macrophages play the most crucial role, secreting cytokines that determine the immune response and modulate the subsequent bone regeneration. Nanomaterials can assist in regulating immune responses by targeting macrophages and managing their polarization, bringing a new strategy for managing bone-related diseases [151].

## 7. Conclusions and Future Remarks

NPs have been widely applied in bone regeneration and showed great potential in osteoimmunomodulation. However, certain disadvantages, such as biocompatibility, immunogenic properties, and toxicity, limit the clinical application of NPs. Additionally, how to ensure the NPs target the bone marrow macrophages instead of macrophages in other organs (e.g., spleen, liver, etc.) remains a challenge for future research. Meanwhile, the complex multi-stage regenerative process of bone healing, the discrepancy or mismatch between the degradation rate of NPs and the growth rate of bone tissues, the problem of regulating the release rate of therapeutic cargo (drugs, factors, or genes), and other limitations still pose obstacles to the application of NPs, which still need further improvement. The fabrication process and approach of nanotopography should be enhanced and optimized to modify the immune response accurately. As previously stated, ordinary materials have imprecise chemical properties that are typically overlooked. The administration must consider the chemical characteristics of the outermost surface. Plasma polymerization is an excellent technique for creating a persistent and non-pinhole biocompatible coating on diverse nanostructures, allowing for specific chemical adjustment of the outermost material, thus achieving precision-tuned bio-physicochemical and biomechanical surface properties. With the development of nanomaterials and material modification approaches, macrophage-targeting nanotherapeutics can ensure the drugs are delivered more precisely to the therapeutic site, therefore allowing for advanced osteoimmunomodulation to improve bone regeneration. Furthermore, the improvement of NP-based drug delivery systems enables the delivery of multiple drugs to target the different stages of bone regeneration. For example, immunomodulatory therapeutics can be released in the early stage of bone healing to ensure the local environment suits bone regeneration. The osteogenic factors can be sequentially released later to boost bone regeneration. Other approaches, such as environmental-responsive releases of immunomodulators and osteogenic factors, can facilitate personalized osteoimmunomodulatory regulation and bone healing.

In summary, this review introduced the importance of osteoimmunology in bone regeneration, the types and current biomedical applications of NPs, the multiple roles of NPs in osteogenesis, and specifically, the significance of NP application on macrophage-targeting osteoimmunomodulation for advanced bone regeneration. Therefore, it is expected that advanced nanotechnology will shed light on bone tissue engineering and facilitate functional bone repair in the future.

## Figures and Tables

**Figure 1 nanomaterials-13-00692-f001:**
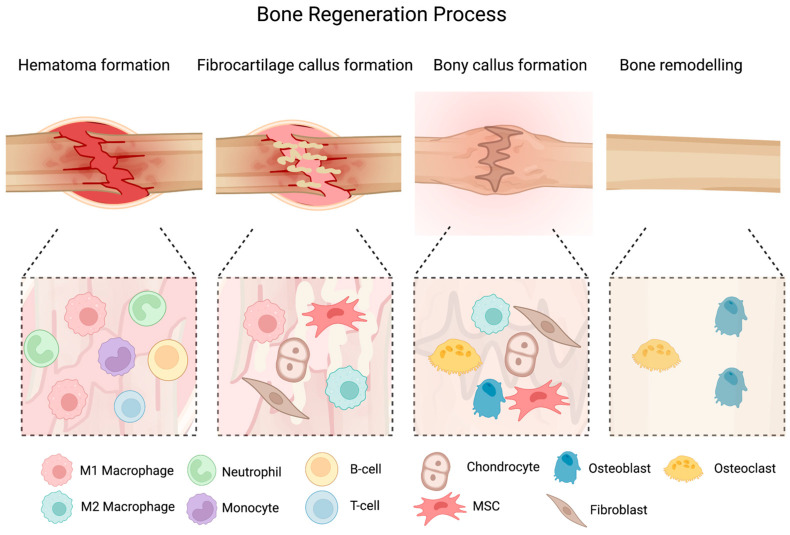
The figure of bone (fracture) healing process. The several phases of bone regeneration are presented, from the initial hematoma formation phase to callus formation and subsequent remodeling. At each phase, the major cell populations are indicated. Created with BioRender.com.

**Figure 2 nanomaterials-13-00692-f002:**
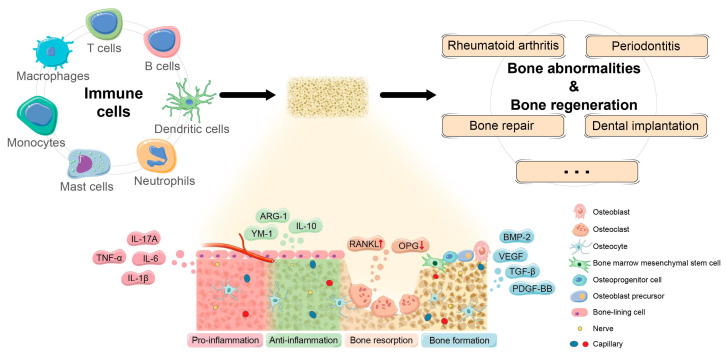
The mechanisms of osteoimmunomodulation in bone regeneration. Reprinted/adapted with permission from Ref. [35]. Copyright 2021 Zhou, Wu, Yu, Tang, Liu, Jia, Yang and Xiang.

**Figure 3 nanomaterials-13-00692-f003:**
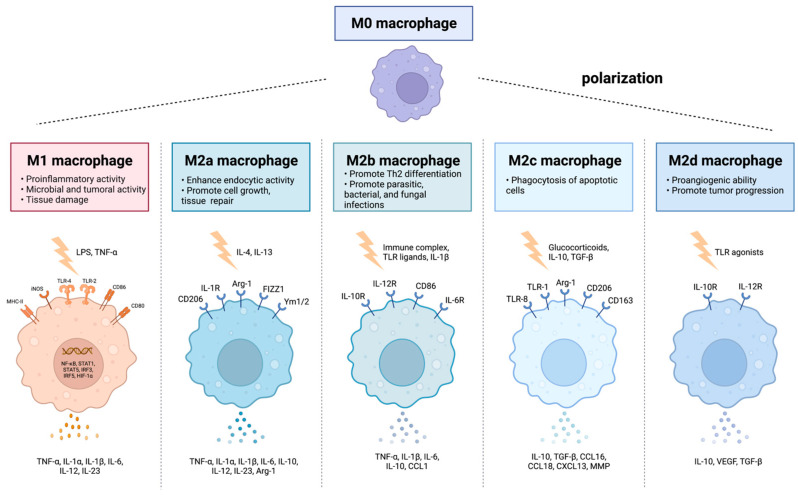
Macrophages have been characterized as unpolarized M0, pro-inflammatory M1 phenotypes, and anti-inflammatory M2 phenotypes (M2a, M2b, and M2c) with different functions. Created with BioRender.com.

**Figure 4 nanomaterials-13-00692-f004:**
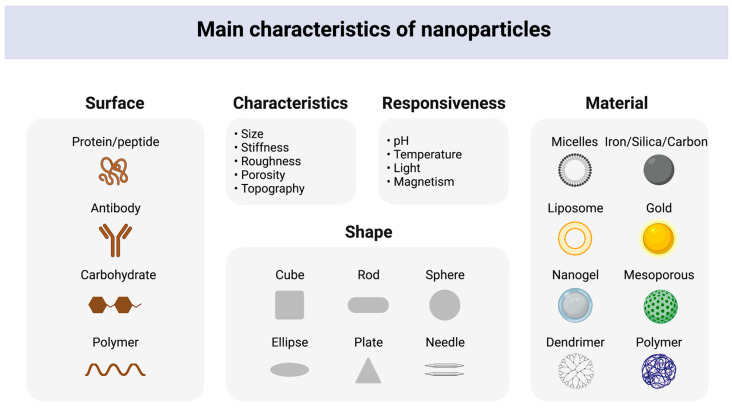
Physical and chemical properties of NPs, including the surface, shapes, characteristics, responsiveness, and nano-based materials. Created with BioRender.com.

**Figure 5 nanomaterials-13-00692-f005:**
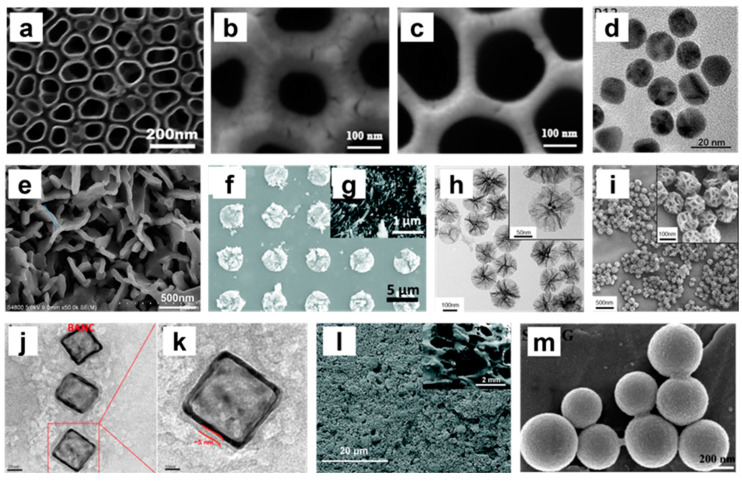
Composite images (TEM and SEM images) of different nanoparticle structures. (**a**) SEM image of 110 nm titania nanotubes (TNTs) [116]; (**b**,**c**) SEM images of anodic alumina structures with different sized pores (100 nm and 200 nm) [119]; (**d**) TEM image of peptide-coated gold NPs (P12) [112]; (**e**) SEM image of surface morphology of SIS/SrHA [150]; (**f**,**g**) SEM images of HA bioceramics with nanoneedle structures. (g: high magnification image) [120]; (**h**) TEM images of 150 nm extra-large pore mesoporous silica NPs (XL-MSNs) (inset: high magnification image) [133]; (**i**) SEM images of 150 nm XL-MSNs (inset: high magnification image) [133]; (**j**,**k**) TEM images of biomimetic anti-inflammatory nano-capsules (BANC) [139]; (**l**) SEM image of 5 wt% ZnO incorporated microcrystalline bioactive glass (5Zn-MCBG) [149]; (**m**) SEM image of strontium-substituted submicrometer bioactive glass (Sr–SBG) [124]. Reprinted/adapted with permission from Ref. [112]. Copyright 2020 Wang, Zhang, Sun, Gao, Xiong, Ma, Liu, Shen, Li and Yang. Ref. [116]. Copyright 2019 Shen, Yu, Ma, Luo, Hu, Li, He, Zhang, Peng and Song. Ref. [119]. Copyright 2017 Chen, Ni, Han, Crawford, Lu, Wei, Chang, Wu and Xiao. Ref. [124]. Copyright 2016 Zhang, Zhao, Huang, Fu, Li and Chen. Ref. [133]. Copyright 2017 Kwon, Cha, Cho, Min, Park, Kang and Kim. Ref. [139]. Copyright 2020 Yin, Zhao, Li, Zhao, Wang, Deng, Zhang, Shen, Li and Zhang. Ref. [149]. Copyright 2021 Bai, Liu, Xu, Ye, Zhou, Berg, Yuan, Li and Xia. Ref. [150]. Copyright 2021 Yang, Zhou, Yu, Yang, Sun, Ji, Xiong and Guo.

**Figure 6 nanomaterials-13-00692-f006:**
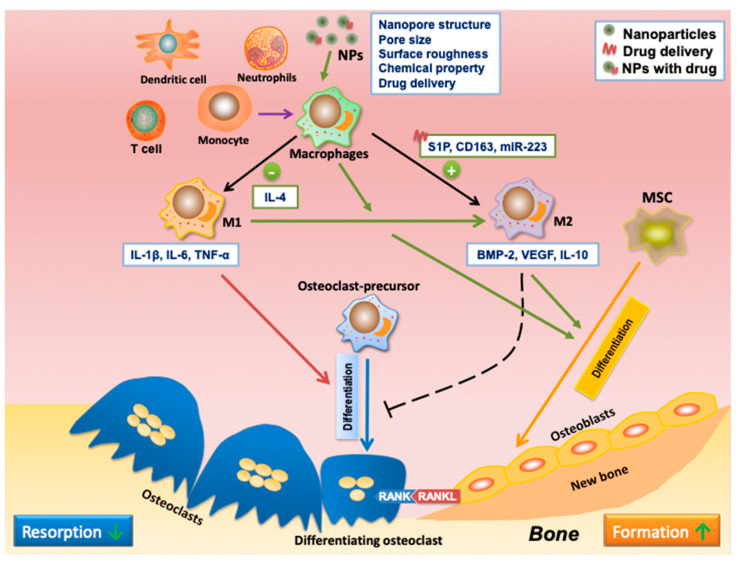
NPs as drug delivery systems to introduce functional osteoimmunomodulation to promote bone regeneration. Ideally, NPs should modulate the immune system to enable the formation of an ideal immune microenvironment for subsequent osteogenesis and bone regeneration. Macrophage polarization is essential in osteoimmunomodulation. The pro-inflammatory M1 phenotype of macrophages could secrete pro-inflammatory cytokines such as IL-1β, IL-6, and TNF-α to promote osteoclastogenesis. The anti-inflammatory M2 phenotype of macrophages could secrete osteogenic cytokines, including BMP2 and VEGF, to enhance bone regeneration. The timely M1-to-M2 phenotype switch is critical in bone regeneration, which can be induced by NP-based drug delivery. NPs can regulate macrophage polarization through different strategies, such as nanopore structure and size, surface roughness, chemical properties, and delivered drugs. NPs can inhibit M1 polarization, promote macrophage polarization to M2, or enhance M1 to M2 polarization, further promoting bone healing.

**Table 1 nanomaterials-13-00692-t001:** Experimental Approach for Osteoimmunomodulation Characterization.

In Vitro and In Vivo Assays	Physical and Chemical Characterization	Biocompatibility Evaluation and Biomechanical Analysis
Cell culture-based assays (osteoblast, osteoclast, macrophage) Enzyme-linked immunosorbent assays (ELISA) Alkaline phosphatase activity assays Mineralization assays	X-ray diffraction (XRD) Transmission electron microscopy (TEM) Scanning electron microscopy (SEM) Dynamic light scattering (DLS)	Cytotoxicity assays (MTT, LDH) Hemocompatibility assays (hemolysis, platelet activation) Inflammation assays (IL-1β, TNF-α, IL-6)
Implantation studies in animal models (rats, mice, rabbits) Histological analysis (bone formation and resorption) Micro-computed tomography (μCT) Bone density measurements	Fourier transform infrared spectroscopy (FTIR) Energy-dispersive X-ray spectroscopy (EDS) X-ray fluorescence (XRF) Nuclear magnetic resonance spectroscopy (NMR)	Contact angle measurements Zeta potential measurements Surface roughness analysis Compression tests Tensile tests Indentation tests

**Table 2 nanomaterials-13-00692-t002:** Applications of NPs in osteoimunomodulation via modulating macrophage response.

Strategies for Regulating Macrophage Polarization	Applications of NPs in Osteoimunomodulation	References
Intrinsic properties	Gold, TiO_2,_ and cerium oxide (CeO_2_) NPs can enhance M2 polarization.	[116,117,118]
Nanopore structure and pore size	NPs with pores of larger size (100 and 200 nm) were highly anti-inflammatory and inhibited M1 polarization.	[119]
	The nanoneedle structure induced M2 polarization. The micropattern sizes of 12 μm and 36 μm in the micro/nano hierarchy enhanced M2 polarization.	[120]
Surface roughness	Ti with smooth surface stimulated M1 activation. Ti with rough surface enhanced M2 polarization.	[122]
Composition	Gold NPs fused hexapeptides Cys-Leu-Pro-Phe-Phe-Asp, peptide arginine-glycine-aspartic acid (RGD), and IL-4 could stimulate M2 polarization.	[112,126,127]
	CeO_2_ NPs with hydroxyapatite could enhance M2 polarization.	[128]
	Strontium (Sr)- or copper (Cu)-doped bioactive glass particles promoted M2 polarization and enhanced osteogenesis.	[124,125]
Drug delivery	Various nanocarriers have delivered IL-4 (anti-inflammatory cytokine) to induce M2 polarization.	[131,132,133]
	NPs can deliver S1P synthetic analog to direct macrophage polarization toward M2.	[134]
	CD163 gene has been encapsulated into polyethyleneimine NPs decorated with a mannose ligand to induce CD163 expression and macrophage polarization toward M2.	[137]
	miR-223 5p mimic was delivered to induce macrophage polarization to M2.	[138]
	Resolvin D1-loaded gold nanocages (AuNC) were coated with M1-like macrophage membranes to enhance M2 polarization.	[139]
A sequential release of therapeutics induces the M1-to-M2 phenotype switch during tissue regeneration.	Spillar et al. designed a scaffold that achieved a sequential release of first IFN-γ and then IL-4 to modulate macrophage polarization from early stage M1 to later-stage M2.	[142]
	NPs carry both miRNA-155 and miRNA-21 to sequentially stimulate macrophage polarization first toward M1 and then the M2 phenotype.	[146]
	Microcrystalline bioactive glass scaffolds with different doses of ZnO orchestrate the sequential M1-to-M2 macrophage polarization.	[149]
	Sr-substituted nanohydroxyapatite (nano-SrHA) coatings and IFN-γ to the surface of native SIS membrane control a sequential M1-M2 macrophage transition.	[150]

## Data Availability

Not applicable.

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
