# Peer review of "Osteoimmunomodulatory Nanoparticles for Bone Regeneration"

_nanomaterials, 2023, doi:10.3390/nano13040692_

Round 1
Reviewer 1 Report
The article fits the journal. However, there are several missing references, especially considering it is a review article.
Baldelli, A., Ou, J., Li, W. and Amirfazli, A., 2020. Spray-on nanocomposite coatings: wettability and conductivity. Langmuir, 36(39), pp.11393-11410.
Zhou, X., Cornel, E.J., He, S. and Du, J., 2021. Recent advances in bone-targeting nanoparticles for biomedical applications. Materials Chemistry Frontiers.
Nirmala, R., Sheikh, F.A., Kanjwal, M.A., Lee, J.H., Park, S.J., Navamathavan, R. and Kim, H.Y., 2011. Synthesis and characterization of bovine femur bone hydroxyapatite containing silver nanoparticles for the biomedical applications. Journal of Nanoparticle Research, 13(5), pp.1917-1927.
Narciso, A.M., da Rosa, C.G., Nunes, M.R., Sganzerla, W.G., Hansen, C.M., de Melo, A.P.Z., Paes, J.V., Bertoldi, F.C., Barreto, P.L.M. and Masiero, A.V., 2021. Antimicrobial green silver nanoparticles in bone grafts functionalization for biomedical applications. Biocatalysis and Agricultural Biotechnology, 35, p.102074.
Wu, Y., Jiang, W., Wen, X., He, B., Zeng, X., Wang, G. and Gu, Z., 2010. A novel calcium phosphate ceramic–magnetic nanoparticle composite as a potential bone substitute. Biomedical Materials, 5(1), p.015001.
and so on...
Besides, there is a lack of the author's suggestion based on their experience.
Author Response
Response to Comments from Reviewer 1
Point 1: The article fits the journal. However, there are several missing references, especially considering it is a review article.
Baldelli, A., Ou, J., Li, W. and Amirfazli, A., 2020. Spray-on nanocomposite coatings: wettability and conductivity. Langmuir, 36(39), pp.11393-11410.
Zhou, X., Cornel, E.J., He, S. and Du, J., 2021. Recent advances in bone-targeting nanoparticles for biomedical applications. Materials Chemistry Frontiers.
Nirmala, R., Sheikh, F.A., Kanjwal, M.A., Lee, J.H., Park, S.J., Navamathavan, R. and Kim, H.Y., 2011. Synthesis and characterization of bovine femur bone hydroxyapatite containing silver nanoparticles for the biomedical applications. Journal of Nanoparticle Research, 13(5), pp.1917-1927.
Narciso, A.M., da Rosa, C.G., Nunes, M.R., Sganzerla, W.G., Hansen, C.M., de Melo, A.P.Z., Paes, J.V., Bertoldi, F.C., Barreto, P.L.M. and Masiero, A.V., 2021. Antimicrobial green silver nanoparticles in bone grafts functionalization for biomedical applications. Biocatalysis and Agricultural Biotechnology, 35, p.102074.
Wu, Y., Jiang, W., Wen, X., He, B., Zeng, X., Wang, G. and Gu, Z., 2010. A novel calcium phosphate ceramic–magnetic nanoparticle composite as a potential bone substitute. Biomedical Materials, 5(1), p.015001.
and so on...
Besides, there is a lack of the author's suggestion based on their experience.
Response 1: We thank the reviewer for this valuable suggestion. The suggested references have been cited in the revised manuscript ([48], [95], [91], [93], [68]). An authors’ suggestion part has been added in the section “Conclusion and future remarks”. Details can be found in Page 15 line 712 of the revised manuscript.

Reviewer 2 Report
This review article introduces the osteoimmunomodulatory effect of skeletal cells such as osteoblasts, osteoclasts, and immune cells to regulate bone formation and resorption. The authors focus on the macrophages that play a crucial role in modulating bone regeneration. In the following sections, applications of nanoparticle-based drug delivery systems that enable the delivery of therapeutic agents into macrophages and the controlled release of therapeutics are reviewed as a potent therapeutic strategy for bone regeneration based on the osteoimmunomodulatory approach. Though this manuscript well reviews the importance of osteoimmunology in bone regeneration and the potential applications of nanoparticles in bone regeneration and osteoimmunomodulation, the description of several points such as targeting factors and biocompatibility is not sufficient to support conclusions. Below are my comments and suggestions that should be addressed before recommending for publication.
1. It has been well-studied that the clearance of nanoparticles from blood circulation is carried by the mononuclear phagocyte system (MPS) including macrophages and Kupffer cells in the spleen, liver, and bone marrow. On this point, nanoparticle-based drug delivery systems would be promising to deliver therapeutics into macrophages as described in the manuscript. However, many studies indicated that MPS in the spleen and liver is a major function for the uptake of circulating nanoparticles. Therefore, the strategy to specifically target bone marrow macrophages while escaping from the uptake by spleen and liver should be a key point for developing osteoimmunomodulatory nanoparticles and therapeutic delivery. Are there any specific receptors on the bone marrow macrophages for the targeted delivery? The strategy to target bone marrow macrophages should be described in more detail.
2. From the viewpoint of comment 1, Table 1 is strategies for regulating macrophage polarization rather than ”Strategies for targeting macrophages”.
3. To support the conclusion that “However, certain disadvantages, such as biocompatibility, immunogenic properties, and toxicity limit the clinical application of NPs.”, biocompatibility, immunogenic properties, and toxicity of nanoparticles should be reviewed in the former sections.
Author Response
Response to Comments from Reviewer 2
Comment: This review article introduces the osteoimmunomodulatory effect of skeletal cells such as osteoblasts, osteoclasts, and immune cells to regulate bone formation and resorption. The authors focus on the macrophages that play a crucial role in modulating bone regeneration. In the following sections, applications of nanoparticle-based drug delivery systems that enable the delivery of therapeutic agents into macrophages and the controlled release of therapeutics are reviewed as a potent therapeutic strategy for bone regeneration based on the osteoimmunomodulatory approach. Though this manuscript well reviews the importance of osteoimmunology in bone regeneration and the potential applications of nanoparticles in bone regeneration and osteoimmunomodulation, the description of several points such as targeting factors and biocompatibility is not sufficient to support conclusions. Below are my comments and suggestions that should be addressed before recommending for publication.
A: We thank the reviewer for these valuable comments.
Point 1: It has been well-studied that the clearance of nanoparticles from blood circulation is carried by the mononuclear phagocyte system (MPS) including macrophages and Kupffer cells in the spleen, liver, and bone marrow. On this point, nanoparticle-based drug delivery systems would be promising to deliver therapeutics into macrophages as described in the manuscript. However, many studies indicated that MPS in the spleen and liver is a major function for the uptake of circulating nanoparticles. Therefore, the strategy to specifically target bone marrow macrophages while escaping from the uptake by spleen and liver should be a key point for developing osteoimmunomodulatory nanoparticles and therapeutic delivery. Are there any specific receptors on the bone marrow macrophages for the targeted delivery? The strategy to target bone marrow macrophages should be described in more detail.
Response 1: We thank the reviewer for this valuable suggestion. Currently it is challenging to target a specific macrophage type (e.g., specifically target the bone marrow macrophages instead of Kupffer cells), which is a limitation in nanoparticle application and has been pointed out in the “Conclusion and future remarks” (Page 15). On the other hand, for bone regeneration, nanomaterials such as nanoparticles are usually used with bone implants (via approaches such as surface coating, 3D printing, etc.) for local drug delivery/immunomodulation, which can target the local bone marrow macrophages instead of macrophage in the spleen and liver. Details regarding this point have been discussed and can be found in Page 12 line 624 and Page 15 line 698 of the revised manuscript.
Point 2: From the viewpoint of comment 1, Table 1 is strategies for regulating macrophage polarization rather than” Strategies for targeting macrophages”.
Response 2: We thank the reviewer for this valuable suggestion and have corrected the name of the table (current Table 2). Details can be found in Page14 line 674.
Point 3: To support the conclusion that “However, certain disadvantages, such as biocompatibility, immunogenic properties, and toxicity limit the clinical application of NPs.”, biocompatibility, immunogenic properties, and toxicity of nanoparticles should be reviewed in the former sections.
Response 2: We thank the reviewer for this valuable suggestion, and have added the contents to discuss the biocompatibility, immunogenic properties, and toxicity of nanoparticles. Details can be found in Page 10 line 453 of the revised manuscript.

Reviewer 3 Report
The authors have submitted an interesting review article " Osteoimmunomodulatory Nanoparticles for Bone Regeneration" which briefly explains the significance of osteoimmunology in bone regeneration and summarizes the advancement and application of NP-based approaches for bone regeneration from an osteoimmunomodulation perspective. The manuscript is well structured and reads well overall, although it will need some revisions. I suggest this article be published after a major revision.
*** General comments:
ü The abstract is clear and concise and comprises all cornerstones including a brief/general introduction to the topic, a non-technical summary of the significant findings, and their implications.
ü The introduction is compelling, clear, and concise. The introduction part covers a proper description of the challenge/gap and a strong background in the field associated with a fair literature review, however, it can be improved further.
ü The various sections of the body of the text are clear and concise overall.
ü The conclusions are logical.
*** Suggested revisions:
1- First of all, I strongly recommend the authors provide a simple, high-quality, and informative “Graphical abstract” which can present the whole concept of your study at a glance. I recommend authors design a “Graphical Abstract” for this study to better show the whole story in a simple and informative manner. In this regard, you can use illustrate a simple sketch of the big picture using the “Biorender” website.
2- Please carefully revise the manuscript to remove grammatical errors and vague sentences. Some of the sentences are unnecessarily long which makes it difficult and boring for the readers to follow them. Please double-check the whole manuscript and revise all.
3- The novelty statement of an article is of significant importance that highlights the importance of the current study and separates it from previously done research. In this work, the novelty statement poorly represents the work and the authors needed more development and better define their hypothesis and objectives and how the presented work differs from already and recently published reports in the field with similar concepts such as:
“Novel insights into nanomaterials for immunomodulatory bone regeneration - https://doi.org/10.1039/D1NA00741F”
4- Some of the references are too old (e.g., 1998, 2000, 2003, etc.). A myriad of research bodies has been published in recent years and you can find similar concepts and cite them in your paper rather than more than 2 decades old references. Moreover, in the introduction part to better present the fundamentals of this field, please read and add valuable information from the following key papers as well:
Introduction section/ Page 2/ lines 67: https://doi.org/10.3390/ijms23042223
5- I suggest the authors provide the readers of this manuscript with a deeper direction of future research of this field either in a distinct section “future perspective”, or along with the remaining challenges in the same part like “remaining challenges and future perspective”.
6- Please ensure you have all the required permission for adapted figures.
Author Response
Response to Comments from Reviewer 3
Comment: The authors have submitted an interesting review article. "Osteoimmunomodulatory Nanoparticles for Bone Regeneration” which briefly explains the significance of osteoimmunology in bone regeneration and summarizes the advancement and application of NP-based approaches for bone regeneration from an osteoimmunomodulation perspective. The manuscript is well structured and reads well overall, although it will need some revisions. I suggest this article be published after a major revision.
*** General comments:
ü The abstract is clear and concise and comprises all cornerstones including a brief/general introduction to the topic, anon-technical summary of the significant findings, and their implications.
ü The introduction is compelling, clear, and concise. The introduction part covers a proper description of the challenge/gap and a strong background in the field associated with a fair literature review, however, it can be improved further.
ü The various sections of the body of the text are clear and concise overall.
ü The conclusions are logical.
A: We thank the reviewer for these positive comments.
Point 1: First of all, I strongly recommend the authors provide a simple, high-quality, and informative “Graphical abstract” which can present the whole concept of your study at a glance. I recommend authors design a “Graphical Abstract” for this study to better show the whole story in a simple and informative manner. In this regard, you can use illustrate a simple sketch of the big picture using the “Biorender” website.
Response 1: We thank the reviewer for this valuable suggestion. Accordingly, a Graphical abstract has been added in Page 1 line 31 of the revised manuscript using Biorender software.
Point 2: Please carefully revise the manuscript to remove grammatical errors and vague sentences. Some of the sentences are unnecessarily long which makes it difficult and boring for the readers to follow them. Please double-check the whole manuscript and revise all.
Response 2: We thank the reviewer for this valuable suggestion and have revised the manuscript to remove grammatical errors and vague sentences.
Point 3: The novelty statement of an article is of significant importance that highlights the importance of the current study and separates it from previously done research. In this work, the novelty statement poorly represents the work and the authors needed more development and better define their hypothesis and objectives and how the presented work differs from already and recently published reports in the field with similar concepts such as:
“Novel insights into nanomaterials for immunomodulatory bone regeneration - https://doi.org/10.1039/D1NA00741F”
Response 3: We thank the reviewer for this valuable suggestion. Accordingly, the innovation and novelty of this review have been highlighted. Details can be found in Page 2 line 79 of the revised manuscript.
Point 4: Some of the references are too old (e.g., 1998, 2000, 2003,etc.). A myriad of research bodies has been published in recent years and you can find similar concepts and cite them in your paper rather than more than 2 decades old references. Moreover, in the introduction part to better present the fundamentals of this field, please read and add valuable information from the following key papers as well:
Introduction section/ Page 2/ lines 67: https://doi.org/10.3390/ijms23042223
Response 4: We thank the reviewer for this valuable suggestion and have removed the old references from the review. Additionally, the key paper has been discussed in the introduction part (Page 2 line 75) of the revised manuscript.
Point 5: I suggest the authors provide the readers of this manuscript with a deeper direction of future research of this field either in a distinct section “future perspective”, or along with the remaining challenges in the same part like “remaining challenges and future perspective”.
Response 5: We thank the reviewer for this valuable suggestion and have discussed the future perspective and remaining challenges in “Conclusion and future remarks”. Details can be found in the Page 15 line 712 of the revised manuscript.
Point 6: Please ensure you have all the required permission for adapted figures.
Response 6: We thank the reviewer for this valuable suggestion. For this review, every figure was drafted by the authors except for Figure 2 and Figure 6, and the permission has been obtained.

Reviewer 4 Report
Authors should include and explain the following comments
1. Physical and chemical properties of nanomaterials and structural effects of nanomaterials
2. Overview of the experimental approaches and characterization methods used to evaluate the osteoimmunomodulatory properties of nanomaterials like polymers, ceramics, composites, metals etc. along with table
3. Osteoimmunomodulatory effects of modification strategies
4. Different categories of osteoimmunomodulatory nanomaterials
5. Involvement of immune cells in bone regeneration
6. Functionalization for osteoimmunomodulation
7. Macrophages in bone regeneration (Scheme or Figure)
Author Response
Response to Comments from Reviewer 4
Point 1: Physical and chemical properties of nanomaterials and structural effects of nanomaterials.
Response 1: We thank the reviewer for this valuable suggestion. The content regarding the physical and chemical properties of nanomaterials and structural effects of nanomaterials has been added in Page 6 line 244 of the revised manuscript.
Point 2: Overview of the experimental approaches and characterization methods used to evaluate the osteoimmunomodulatory properties of nanomaterials like polymers, ceramics, composites, metals etc. along with table.
Response 2: We thank the reviewer for this valuable suggestion. We have briefly described the experimental approaches and characterization methods, and a table (Table 1) was added to summarize them. Details can be found in the Page 10 line 482 and Page 13 line 668 of the revised manuscript.
Point 3: Osteoimmunomodulatory effects of modification strategies.
Response 3: We thank the reviewer for this valuable suggestion, based on which, the osteoimmunomodulatory effects of the modification strategies has been discussed in the Chapter 6. Details can be found in Page 11 line 538 of the revised manuscript.
Point 4: Different categories of osteoimmunomodulatory nanomaterials.
Response 4: We thank the reviewer for this valuable suggestion. The content regarding the categories of osteoimmunomodulatory nanomaterials has been added accordingly. Details can be found in Page 6 line 261 of the revised manuscript.
Point 5: Involvement of immune cells in bone regeneration.
Response 5: We thank the reviewer for this valuable suggestion. The roles of immune cells in bone regeneration have been discussed in the chapter 3, and we have added the overview of the involvement of immune cells in bone regeneration. Details can be found in the Page 4 line 160 of the revised manuscript.
Point 6: Functionalization for osteoimmunomodulation.
Response 6: We thank the reviewer for this valuable suggestion. The functionalization approaches for osteoimmunomodulatory materials has been discussed in Chapter 6. Details can be found in the Page 10 line 475 of the revised manuscript.
Point 7: Macrophages in bone regeneration (Scheme or Figure).
Response 6: We thank the reviewer for this valuable suggestion, based on which the Figure 2, Figure 3, Figure 5 have been added to describe the role of macrophages in bone regeneration. Details can be found in the Page 5 and Page 13 of the revised manuscript.

Reviewer 5 Report
The manuscript submitted to Nanomaterials entitled "Osteoimmunomodulatory Nanoparticles for Bone Regeneration" by Jingyi Wen et al. presents a concise review with a simple and well-organized structure. The authors carefully discuss the subject, and I don't have any concerns about its quality and importance. However, only two illustrations are made in the entire revision. At least one image per section should be made. It's much simpler to understand complex scientific mechanisms with figures than reading paragraphs.
Author Response
Response to Comments from Reviewer 5
Point 1: The manuscript submitted to Nanomaterials entitled "Osteoimmunomodulatory Nanoparticles for Bone Regeneration" by Jingyi Wen et al. presents a concise review with a simple and well-organized structure. The authors carefully discuss the subject, and I don't have any concerns about its quality and importance. However, only two illustrations are made in the entire revision. At least one image per section should be made. It's much simpler to understand complex scientific mechanisms with figures than reading paragraphs.
Response 1: We thank the reviewer for these positive comments and valuable suggestions. We have added the figure for each section of this review. Details can be found in the Page 3, 5, 7 and Page 13 of the revised manuscript.

Round 2
Reviewer 1 Report
Great improvements. I think it would be ready to go.
Reviewer 3 Report
The manuscript is well-amended and the authors well answered all my concerns. I have no further comments.
Reviewer 4 Report
Accept in present form